# Fecal Microbiota Transplantation in the Treatment of Chronic Pouchitis: A Systematic Review

**DOI:** 10.3390/microorganisms8091433

**Published:** 2020-09-18

**Authors:** Frederik Cold, Sabrina Just Kousgaard, Sofie Ingdam Halkjaer, Andreas Munk Petersen, Hans Linde Nielsen, Ole Thorlacius-Ussing, Lars Hestbjerg Hansen

**Affiliations:** 1Department of Plant and Environmental Sciences, Section for Microbial Ecology and Biotechnology, Copenhagen University, Thorvaldsensvej 40, 1871 Frederiksberg C, Denmark; frederik.cold@regionh.dk; 2Gastrounit, Medical Division, Copenhagen University Hospital Hvidovre, 2650 Hvidovre, Denmark; sofie.ingdam.halkjaer@regionh.dk (S.I.H.); andreas.munk.petersen@regionh.dk (A.M.P.); 3Department of Gastrointestinal Surgery, Aalborg University Hospital, 9100 Aalborg, Denmark; s.kousgaard@rn.dk (S.J.K.); otu@rn.dk (O.T.-U.); 4Department of Clinical Medicine, Aalborg University, 9100 Aalborg, Denmark; halin@rn.dk; 5Department of Clinical Microbiology, Copenhagen University Hospital Hvidovre, 2650 Hvidovre, Denmark; 6Department of Clinical Microbiology, Aalborg University Hospital, 9100 Aalborg, Denmark

**Keywords:** pouchitis, fecal microbiota transplantation, microbiota, IPAA

## Abstract

The objective was to evaluate available literature on treatment of chronic pouchitis with fecal microbiota transplantation (FMT) focusing on clinical outcomes, safety, and different approaches to FMT preparation and delivery. A systematic review of electronic databases was conducted using Medline, EMBASE, and the Cochrane Central Register of Controlled Trials Library from inception through April 2020. Human studies of all study types reporting results of FMT to treat chronic pouchitis were included. Nine studies, reporting FMT treatment of 69 patients with chronic pouchitis were found eligible for the review. Most studies were case series and cohort studies rated as having fair to poor quality due to high risk of bias and small sample size. Only one randomized controlled trial was included, finding no beneficial effect of FMT. In total clinical response after FMT was reported in 14 (31.8%) out of 44 evaluated patients at various timepoints after FMT, and clinical remission in ten (22.7%) patients. Only minor self-limiting adverse events were reported. FMT varied greatly regarding preparation, length of treatment, and route of delivery. The effects of FMT on symptoms of chronic pouchitis are not established, though some studies show promising results. Future controlled well-designed studies are warranted.

## 1. Introduction

Restorative proctocolectomy with ileal pouch anal anastomosis (IPAA) is the surgical treatment in patients with ulcerative colitis (UC) refractory to medicinal therapy, in some cases of colorectal cancer and in familial adenomatous polyposis [1]. However, pouchitis occurs in up to 60% of the UC patients after surgery [2,3]. Pouchitis is characterized by inflammation, mainly confined to the pouch, with symptoms such as increased bowel movements, fever, bloody stool, fatigue, and abdominal pain [2,4]. Pouchitis often develops as an acute, but transient, inflammation responding to antibiotics. For 5% of patients, the inflammation becomes chronic, being antibiotic-dependent, with the need of continuously antibiotic treatment, or refractory, not responding to the standard treatment with antibiotics [5,6,7].

The etiology of pouchitis remains unclear, but the gut microbiome is hypothesized as a key factor. The pouch microbiome of patients with pouchitis is different compared to patients with a non-inflamed pouch [8,9]. The dysbiotic microbiome of an acute or chronic inflamed pouch is characterized by lowered bacterial diversity and changed abundance of certain bacteria [8,10,11,12]. Antibiotics can in many cases relieve symptoms of pouchitis and the use of probiotics can decrease the risk of developing pouchitis, further indicating the significance of the gut microbiome in pouchitis [12,13,14].

Pouchitis is usually treated with ciprofloxacin and/or metronidazole, which in case of chronic pouchitis often fails [15]. Treatment of chronic pouchitis is challenging with limited therapeutic options, potentially leading to the need for biological treatment or surgical removal of the ileal pouch [5,6,7].

Modification of the fecal microbiome in patients with chronic pouchitis has received increased attention within recent years [16]. The positive results of treating recurrent *Clostridioides difficile* infection (rCDI), with fecal microbiota transplantation (FMT) [17], and the promising results in inflammatory bowel disease (IBD) [18,19], has led to the interest of inverting the gut dysbiosis of patients with chronic pouchitis using FMT to potentially relieve symptoms. 

Within recent years, several smaller studies have reported both positive and negative clinical results of treating chronic pouchitis with FMT [20,21,22,23].

This systematic review aims to review the current literature on treatment of chronic pouchitis with FMT focusing on clinical effects, safety and the different approaches to FMT treatment.

## 2. Methods

### 2.1. Search Strategy and Study Selection

This systematic review was performed in accordance with the PRISMA 2009 guidelines [24]. A literature search was performed using Medline (from 1948), EMBASE (from 1947), and the Cochrane Central Register of Controlled Trials Library (for all years) through 15 April 2020. Bibliographies of review articles and meta-analyses were searched to identify additional studies [8,25,26]. Furthermore, the bibliography of the primary author of the included studies were reviewed to search for further eligible publications. Web of Science [27] was used to identify additional potential publications that have cited the included studies. Clinicaltrials.gov [28] and WHO International Clinical Trials Registry Platform [29] were used to search for results from unpublished studies. To search for grey literature opengrey.eu [30] was used. The detailed search strategy is outlined in the online Appendix A.

Eligibility criteria for inclusion of studies were defined prior to the search through registration of the research protocol in Prospero International Prospective Registry of Systematic Reviews (CRD42020167258) [31]. Inclusion criteria were “human interventional studies using FMT of all study types”, including randomized controlled trials, non-randomized controlled studies, cohort studies, and case studies (case series and case reports), to “treat chronic pouchitis” (recurrent or antibiotic refractory) “reporting clinical outcomes on pouch symptoms”. Studies including “participants of all ages” were included. The search was restricted to studies written in or translated to English. In controlled studies, the accepted comparator was either placebo, autologous FMT, or no treatment. Data presented as a conference abstract were also accepted. Exclusion criteria were studies where FMT was given as primary treatment to treat other conditions than chronic pouchitis. 

All titles and abstracts from the literature search were screened for potential eligibility by two investigators (FC and SJK) independently, and in strict accordance with the inclusion and exclusion criteria. In case of dispute, the key decision was made by a third investigator (AMP).

### 2.2. Data Management and Analysis

Data extraction was performed independently by two investigators (FC and SJK). The following clinical information was extracted from each included study, if present: first author, year of publication, study location, study design, age and characteristics of study population, definition of condition under consideration including severity, details of intervention and methodology (such as dosage, frequency, route of administration, duration, and preparation of FMT material), primary and secondary outcome measures and results, duration of follow-up, change in the microbiome after FMT, registered adverse events (deaths, hospital admissions, and other adverse events as defined in each of the included studies), and donor characteristics.

The primary outcome was change in symptoms related to pouchitis compared to before FMT using the pouchitis disease activity index (PDAI) [32]. The definitions of clinical response (reduction in PDAI ≥3) and clinical remission (reduction in PDAI ≥3 and total PDAI of <7) were used to evaluate efficacy from studies reporting PDAI scores [7,33]. The use of the modified PDAI (mPDAI) and clinical PDAI (cPDAI) was accepted [34]. It was accepted that the evaluation of symptoms was performed at different timepoints after FMT.

Data was extracted as an intention-to-treat analysis, with dropouts assumed to be treatment failures. In case of missing data or need for further clarification, the corresponding author of the included study was contacted to retrieve further information. For the primary outcome of clinical response and remission, pooled estimates were calculated for all patients treated with FMT. 

### 2.3. Risk of Bias and Quality Assessment

The Cochrane risk of bias tool was used to assess for bias in randomized controlled trials (RCTs) [35]. Risk of bias in the cohort and case studies was assessed using the US National Heart, Lung, and Blood Institute quality assessment tool [36], which has been used in other systematic reviews of FMT treatment [37], with four weeks selected as cut-off for appropriate follow-up. Further description of assessment of risk of bias and quality is available in the online Appendix A.

## 3. Results

The initial literature search identified 892 studies. After the removal of duplicates, 718 studies underwent title and abstract screening. A total of 49 studies were full-text reviewed for eligibility. Of these 40 were excluded for various reasons resulting in nine studies included in the qualitative and quantitative synthesis (Figure 1).

### 3.1. Patient Characteristics and Study Types

A total of 65 patients were treated with FMT in the nine included studies (Table 1). Most of the included studies were case series, case reports or pilot studies. Only one clinical trial by Herfarth et al. [23], randomized and blinded patients to receive either FMT or placebo. The definition of chronic pouchitis varied among studies with no consistency (Table 2). Some studies used certain values of PDAI [33,38] or mPDAI prior to inclusion [23], while other focused on recurrent need for antibiotics (Table 2) [20,22,39].

### 3.2. Clinical Effects of FMT Treatment

The only included RCT, by Herfarth et al. [23], was prematurely stopped because of lower than expected clinical remission rate and low donor engraftment. All of the four FMT treated patients and the two placebo treated patients enrolled failed to respond and needed antibiotic rescue therapy after study treatment. One patient (primarily receiving placebo) of the five patients treated in the following open-label FMT extension phase achieved antibiotic free clinical remission.

The nine included studies did not evaluate the clinical effect of FMT at the same timepoint or used the same methods to evaluate disease activity, why it was not possible to perform a proper meta-analysis. One study did not evaluate treatment response through PDAI in all patients [40] and one study used Fecal Calprotectin (F-Calprotectin) [20]. Hence, pooled estimates of the primary endpoints of clinical response and remission could only be evaluated in 44 patients. Clinical response was achieved in 14 (31.8%) out of 44 evaluated patients at various timepoints after FMT treatment, while clinical remission was achieved in 10 (22.7%) out of 44 evaluated patients at various timepoints after FMT.

Four of the included studies reported endoscopic and histologic PDAI (ePDAI and hPDAI) with a trend of small improvements in two of the studies [22,39], while no change or minor beneficial changes were observed in the other two (Table 2) [38,40]. 

Some studies also evaluated specific symptoms. Selvig et al. [40], reported improvement in number of bowel movements and abdominal pain, while Schmid et al. [21], reported transiently improved bloating and pain with stool urgency and frequency remaining grossly unchanged. 

F-Calprotectin is routinely used as a marker of disease activity in patients with IBD and pouchitis and is found to be correlated to change in symptoms [41]. In the studies where the patients’ symptoms/PDAI improved, F-Calprotectin also decreased [20,22]. F-Calprotectin was stable or decreased insignificantly in the studies finding no or minor beneficial effects after treatment [23,40].

### 3.3. Safety/Adverse Events

No deaths, hospital admissions or serious adverse events considered related to FMT were reported. Minor self-limiting adverse events were reported in several studies [22,38,39,40]. Most of these were gastrointestinal (nausea, abdominal pain, or bloating), but fever, dizziness, fatigue and feeling uncomfortable were also reported (Table 2).

### 3.4. Microbiome Changes after FMT

Six studies assessed if bacterial alpha diversity increased in fecal or mucosal samples after FMT treatment. In five studies, no significant changes were observed [23,33,38,39,40]. This was true in studies using both single and multiple FMT treatments [33,38,39,40]. In the study by Herfarth et al., increased fecal bacterial diversity after FMT was concluded to be mainly caused by the cessation of antibiotic treatment 24 h prior to FMT [23]. Steube et al., found a significant increased bacterial alpha diversity in patients with improved clinical outcomes after FMT treatment [20]. However, an increased bacterial alpha diversity was not necessarily correlated to improved clinical outcomes, and some patients increased their diversity without clinical improvement. 

Seven studies analysed whether the recipients’ microbiota resembled the donors’ microbiota after treatment [20,22,23,33,38,39,40]. Interestingly, increased resemblance to the donors’ microbiota after treatment was in three studies correlated to beneficial clinical effect with significant changes seen only in patient who improved clinically [22,23,40]. 

Four studies reported changed abundance of certain specific bacteria after FMT [20,22,38,40]. Species such as Ruminococcaceae and Lachnospiraceae family and genus *Faecaelibacterium* were enriched as sign of engraftment and others such as *Escherichia coli* decreased in abundance in recipients after FMT treatment. None of these studies reported a correlation to clinical effect of engraftment or decreased abundance of specific bacteria.

### 3.5. Preparation and Delivery of FMT Material

Most of the included studies used FMT material from unrelated donors [20,22,23,38,40], but in three studies FMT material from relatives were used (Table 3) [21,33,38]. Only the study by Kousgaard et al., used FMT from multiple donors [39]. The delivered FMT material varied greatly in the preparation method and delivery approach among the included studies (Table 3). Five studies used fresh fecal material delivered 4–6 h prior to treatment [21,22,33,38], while other used frozen material [22,23,39,40]. The FMT material was delivered to the recipients through both upper [20,22,38] and lower endoscopically delivery [21,23,33,40,42], and delivery through capsules [20,23], or enemas [39] (Table 2). The use of different types of bowel cleansing prior to FMT was reported in several of the studies delivering FMT through lower endoscopy [21,33,40].

The length of FMT treatment also varied between the studies from a single FMT [33,38,42], to up to two weeks of daily treatment [23,39]. Hence, the total delivered FMT dose varied in between the studies. The lowest amount of 30 mL fecal–saline solution was reported in the study by, Landy et al. [38], and the highest amount of a total of 1925 mL FMT material derived from 525 g stool delivered to one of the patients during seven FMTs was reported in the study by Stallmach et al. [22].

Two studies reported stopping antibiotic treatment 24–48 h prior to FMT treatment [23,42], while seven of the 18 patients in the study by Selvig et al. [40] were pre-treated with rifaximin for a total of five days, beginning eight days prior to FMT. The patients in the included studies were in general permitted to continue concomitant treatment while receiving FMT including biological treatment, [40] while antibiotics and probiotics were not permitted [39,42].

### 3.6. Quality Assessment of Studies and Risk Of Bias

Six studies were rated as having fair to poor quality due to high risk of bias and small sample size. Only one RCT, by Herfarth et al. [23], with six included patients was included. Most of the other studies were case series and case reports with no predefined treatment outcomes or inclusion criteria (online Appendix A).

## 4. Discussion

### 4.1. Main Findings

Only nine studies have investigated the effects of FMT on pouchitis symptoms in a total of 65 patients with chronic pouchitis. There is a scarceness of high-quality studies; hence, the true effect of the treatment remains unclear. The only RCT included in this systematic review [23], was prematurely stopped because of low clinical efficacy and low donor microbial engraftment. Some of the included studies reported high response and remission rates [20,22], while others found no or minor beneficial effect of the treatment [21,23,38,40]. The calculated proportions of patients achieving clinical response and remission of 31.8% and 22.7%, respectively, must be interpreted with precaution since a great heterogeneity between the studies exist and the low quality of evidence in most studies. Hence, this systematic review gives no clear answers of whether FMT is beneficial in patients with chronic pouchitis, but points towards the need for new well-designed controlled studies.

### 4.2. Strengths and Limitations

This systematic review is the first to collect data on clinical outcomes from all published studies treating chronic pouchitis patients with FMT. A systematic review on this topic is important, as studies on FMT to chronic pouchitis only have included a very limited number of patients each. The review gives the first full overview of both the clinical indications of treatment, the different treatment approaches, the clinical efficacy, safety of treatment and microbiome changes caused by the treatment.

There are also limitations. The quality of evidence in most of the included studies were evaluated as fair or poor, and only one RCT was included. Furthermore, there was great differences in definition of disease activity both prior to and after treatment. The timepoint where the effect of treatment was assessed also varied among studies. Further, all included studies had different approaches to preparation, delivery and length of FMT treatment, which make interpretation of the overall effects and approaches of treatment very difficult.

### 4.3. Clinical Efficacy of Treatment

Some of the included studies found improved clinical outcomes with decreased symptom scores [22,39], and lowered F-Calprotectin [20], which indicate that some patients with chronic pouchitis may benefit from treatment with FMT. Furthermore, Lan et al. reported potential beneficial effects of FMT for treating *Clostridioides difficile* infection in patients with ileal pouches [43]. The authors reported that FMT treatment of rCDI in patient with ileal pouches, also improved pouchitis symptoms besides the elimination of rCDI symptoms. A further indication of a potential beneficial effect is the finding of a correlation between engraftment of donor microbiome and improved clinical outcomes in the recipients, reported in several of the studies included in the review [20,22,23,40]. Several studies found a missing or low clinical efficacy in the majority of the patients [21,23,38,40], indicating that not all patients might benefit from the FMT treatment or that the treatment should be delivered differently. 

In UC, where gut dysbiosis also has been linked to disease activity [44], FMT treatment has been found superior to placebo treatment in recent RCTs, although the treatment protocols varied in between the studies [18,45,46,47]. The pooled clinical response and remission rates of the RCTs of 49% and 28%, respectively [48], also indicate that not all UC patients benefit from the treatment. The focus is now to predict which patients that may benefit from FMT treatment in both UC and other conditions [49,50].

The general definition of chronic pouchitis is confirmed pouchitis by PDAI with more than four weeks disease activity [7]. In this review, we choose to get a full overview of the FMT treatment and treatment indications. Hence, we included both studies that used antibiotic refractory/dependent pouchitis and several episodes of pouchitis within the last year as inclusion criteria. This makes it almost impossible to transfer conclusions regarding treatment efficacy to all patients with chronic pouchitis. Furthermore, not all studies evaluated the patients with a full PDAI score [20,23,42]. The PDAI score is an 18-point index to assess pouchitis activity, based on clinical symptoms, endoscopically and histologically evaluation and is the most commonly used tool to evaluate pouchitis disease activity [7,51]. The use of the modified scores mPDAI [34] (only including clinical symptoms and endoscopic evaluation) and cPDAI (only including clinical symptoms), makes it easier to include patients in studies because endoscopies and biopsies are not needed, but decreases the comparability of the results. Therefore, we urge that future studies use a full PDAI score before and after treatment with FMT. We also recommend that the commonly used definitions of clinical response (reduction in PDAI ≥3) and remission (reduction in PDAI ≥3 and PDAI of <7) are used in future studies [7].

Further we recommend that evaluation of the patients should be after four weeks, since a beneficial clinical effect of FMT treatment has been reported at this timepoint in several of the included studies [38,39,52].

### 4.4. Safety

Only self-limiting transient mainly gastrointestinal adverse events were reported as related to FMT in the included studies, which are in line with data from FMT treatment to other indications [53,54]. In general, FMT treatment is considered safe, when the current international recommendations considering donor screening are followed [53,55]. A recent death of an immunocompromised patients announced in June 2019, by the Federal Drug Agency (FDA) in the Unites States, following treatment with FMT capsules, where extended-spectrum beta-lactamase (ESBL)-producing *Escherichia coli* were transferred from the donor to the recipient, indicates that donor screening protocols must be elaborate to avoid the transfer of multi-drug resistant microorganisms [56]. No cases of transferred diseases were reported in any of the included studies. The long-term safety of FMT is not fully investigated. Results from patients with rCDI treated with FMT, which are often multi-morbid and frail, does not indicate long-term adverse events [54,57]. FMT to chronic pouchitis possibly requires several FMT treatments to achieve clinical remission similar to UC [18,45,46,48,58], whereas usually one FMT is sufficient to treat rCDI [59]. It is therefore of great importance that future studies investigate the long-term adverse events after FMT to patients with chronic pouchitis.

### 4.5. Microbiome Changes

Whether the changed gut microbiome of patients with chronic pouchitis is a cause or a consequence of the disease is not fully understood [8]. Furthermore, patients with chronic pouchitis often have received several antibiotic treatments, which also changes the microbiota of the pouch [12]. The results from some of the included studies using 16 S sequencing indicate that there is a correlation between improved clinical outcomes and changes in the gut microbiome with increased bacterial alpha diversity or increased resemblance to the healthy donor microbiota after FMT [20,22,23,40], while Herfarth et al. [23], terminated their RCT because of low clinical efficacy and low donor engraftment.

In cases of treating rCDI and inducing clinical remission in UC the following microbiome changes have been correlated to success of treatment after FMT; increased bacterial diversity [60], increased resemblance of the recipients microbiome to the donors [46], and changes in the amount of certain potentially harmful or beneficial bacteria [61]. Hence, it makes sense to design FMT interventions in doses appropriate for such microbiome changes in future trials of chronic pouchitis.

The role of other parts of the gut microbiome than bacteria in chronic pouchitis has not been reported [8]. Knowledge from other diseases indicates the important roles of fungi and bacteriophages in correlation to the effect of FMT [62,63]. Thus, future trials investigating the effect of FMT on other parts of the gut microbiome are important to increase the understanding of the correlation between the microbiome and chronic pouchitis. Furthermore, none of the included studies used next-generation sequencing techniques, such as shotgun metagenomics [64], which are being increasingly used in the evaluation of the effects of FMT in the treatment of other conditions [65]. We recommend future studies to use deeper microbiome analysis, not limited to 16 S sequencing, to advance our understanding of the changes in the microbiome caused by FMT.

### 4.6. FMT Material: Donors

Results from studies using FMT treatment to UC and irritable bowel syndrome (IBS) have indicated that certain donors are more "superior" than others to induce beneficial clinical effects in patients [46,49,66]. The gut dysbiosis of chronic pouchitis is probably different from the gut dysbiosis associated to other conditions: hence, a good donor in treatment of chronic pouchitis may be different than for other conditions [8]. Neither in chronic pouchitis or other diseases there is a full understanding of what defines an ideal donor, though the understanding is increasing [60].

In the included RCT by Herfarth et al. [23], which found no beneficial effect of FMT, a donor delivering stools with a high butyrate content was used. Butyrate has been correlated to high microbial diversity and better therapeutic response in FMT trials treating other conditions [67,68,69,70]. Three of the included studies used a close relative as stool donor [21,33,38], which is not considered inferior, but with the establishment of stool banks, treatment with stool from close relatives will probably become less prevalent [71]. Patients with a pouch without episodes of pouchitis possibly have a healthy microbiome preventing pouchitis and could maybe be ideal donors.

We encourage future studies to continue the assessment of whether certain characteristics of the donor microbiome are associated with beneficial effects. 

### 4.7. FMT Material: Preparation and Delivery

In general, the included studies varied in both the preparation of FMT material, amount of stool used for treatment, and whether the treatment was given shortly after donation or if frozen stored FMT material was used. In the treatment of UC and IBS, where a dysbiotic gut microbiome probably should be changed to a eubiotic through engraftment of beneficial microorganisms from healthy donors, higher doses or multiple deliveries are more effective than lower doses or single FMT treatment [48,66]. We propose future trials to use multiple FMT treatment based upon this knowledge.

### 4.8. FMT Material: Route of Administration

In spite of the anatomic location of the diseased tissue in the lower gastrointestinal tract of chronic pouchitis, several of the included studies used upper administration of FMT material through either capsules [20,23], or upper endoscopy [20,22]. The fact that some of these studies found positive clinical effects indicate that both upper and lower administration may have beneficial effect. For other conditions located in the lower part of the gastrointestinal tract, such as IBS and UC, a treatment with FMT delivered through upper administration has also been found successful [72,73]. Future studies should investigate if a certain route is more beneficial than others. 

### 4.9. Perspectives

As FMT treatment has received a more widespread use and the establishment of stool banks are emerging in many countries [74,75], we hope that more well-designed controlled trials investigating the effects of FMT in chronic pouchitis will be performed. This in spite of the disappointing clinical results from the only RCT published by Herfarth et al. in 2019 [23], possibly due to low donor microbial engraftment. Chronic pouchitis patients have severely impaired health related quality of life [76], and there is strong need of new treatment options. Future well-designed controlled trials using clearly defined disease definitions, comparing the effects of FMT treatment with either placebo, FMT delivered through another route or FMT with a different dose or duration, will be necessary to move the understanding of the true effect of FMT treatment in chronic pouchitis forward. 

## 5. Conclusions

The effects of fecal microbiota transplantation on symptoms of chronic pouchitis are not established, though some studies show promising results. The treatment appears safe, when guidelines for donor screening are followed. Future controlled studies investigating safety, dose, duration, preparation of FMT, administration route, and concomitant treatments are needed to establish whether FMT can be a part of the treatment of chronic pouchitis.

## Figures and Tables

**Figure 1 microorganisms-08-01433-f001:**
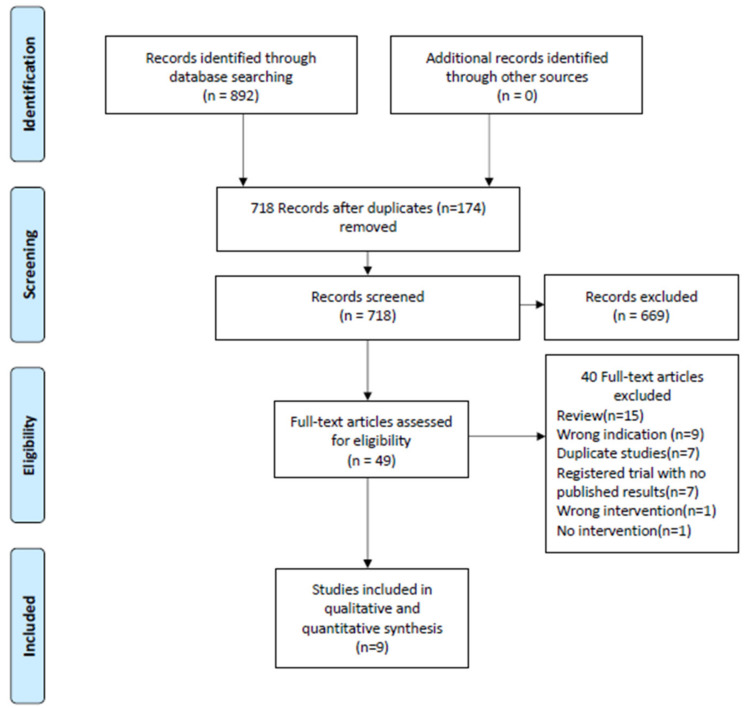
PRISMA flow diagram of assessment of studies identified in the systematic review of fecal microbiota transplantation (FMT) to treat chronic pouchitis.

**Table 1 microorganisms-08-01433-t001:** Patient characteristics, preparation of FMT and different treatment approaches of patients treated with FMT for chronic pouchitis.

Total number of studies, *n*	9
Total patient population, *n*	65
Mean days of FMT treatment, *days* (range)	4.8 (1–14)
Mean delivered amount of stool by FMT, *grams* (range) (*n* = 51)	111.8 (11–525)
Mean follow-up, *days* (range) (*n* = 65)	87.6 (28–365)
Male/Female patients, *n* (*n* = 51)	22/29
Mean age of patients, *years* (range) (*n* = 51)	43.8 (22–77)
Time since restorative proctocolectomy, *mean years* (range) (*n* = 50)	10.3 (1–33)
Single/Multi donor FMT, *n* (*n* = 65)	56/9
Related/Unrelated donor, *n* (*n* = 65)	12/53
Upper/Lower administration/Both, *n* (*n* = 51)	13/32/6
Study type. Patients in RCT/Patients in non-RCT, *n* (*n* = 65)	6/59

FMT, fecal microbiota transplantation; RCT, Randomized Clinical Trial; *n*, number.

**Table 2 microorganisms-08-01433-t002:** Clinical effects of FMT treatment to patients with chronic pouchitis.

Author and Year	Study Type	Patient Population	Sample Size	FMT Treatment	Route	Dosage	Clinical Response (Reduction in PDAI ≥3)	Clinical Remission (Reduction in PDAI ≥3 and Total of <7)	Endoscopic Outcomes	Histologic Outcomes	Adverse Events	Follow-Up
Fang et al., 2016	Case report	Chronic antibiotic resistant pouchitis	1	Single FMT	Sigmoidoscopy (delivered 40 cm into the afferent limb and pouch body)	Stool mixed with saline and diluted to 250 mL	1/1 after 3 months (cPDAI decreased from 6 at baseline to 0)	1/1 after 3 months (cPDAI decreased from 6 at baseline to 0)	NA	NA	No reported adverse events.	3 months
Herfarth et al., 2019	RCT with open-label follow-up	Chronic antibiotic dependent pouchitis >4 weeks. mPDAI ≥5	6 (FMT (4) Placebo (2)). 5 received open-label FMT afterwards	Single endoscopic FMT followed by daily oral encapsulated FMT for 2 weeks	Sigmoidoscopy and oral capsules	eFMT (2 × 30 mL, total of 24 g donor stool) and 6 capsules daily consisting of 4.2 g donor stool	1/6 (Four patients receiving primary FMT and two patients receiving open-label FMT included)	1/6 (clinical PDAI 1 and no need for antibiotics)	NA	NA	No FMT related safety events were observed.	16 weeks
Kousgaard et al., 2020	Cohort (open-label pilot study)	Chronic pouchitis (≥3 episodes of pouchitis within the last year)	9	14 days of daily self-administered FMT	Enema	20 g fecal material diluted in 100 mL saline	3/9 after four weeks	3/9 after four weeks	Mean ePDAI of 3.2 at baseline decreased to 2.2 after four weeks	Mean hPDAI of 1.7 decreased to 1.0 after four weeks	7/9 patients reported adverse events while treated. Abdominal pain (5), uncomfortable (2), nausea (2), fever (2), bloating (1), dizziness (1), and fatigue (1).	6 months
Landy et al., 2015	Cohort (pilot study)	Chronic pouchitis with current PDAI ≥ 7	8	Single FMT	Nasogastric	30 mL of fecal-saline solution followed by 50 mL saline	2/8 after 4 weeks	0/8 after 4 weeks	Mean ePDAI of 5 at both baseline and after 4 weeks	Mean hPDAI of 3 at baseline decreased to 2 after four weeks	Nausea (3), bloating (2), vomiting (1), fever (1). All transient (<24 h).	4 weeks
Nishida et al., 2019	Case series	Chronic pouchitis with current PDAI ≥7	3	Single FMT	Colonoscopy	150–200 g donor stool mixed with 350–500 mL sterile saline	1/3 after 8 weeks	0/3 after 8 weeks	NA	NA	No reported adverse events.	8 weeks
Schmid et al., 2017	Case report	A severe flare of pouchitis in a patient diagnosed with pouchitis one year earlier	1	A total of three FMTs at baseline, after 5 and 9 weeks	Pouchoscopy	250 mL fecal-saline suspension	0/1 after 9 weeks	0/1 after 9 weeks	NA	NA	No reported adverse events.	9 weeks
Selvig et al., 2020	Cohort (prospective open-label pilot study)	Chronic pouchitis. Prior endoscopic evaluation confirming inflammation and over 4 weeks of symptoms	18 (7 of the 18 patients pre-treated with Rifaximin)	1 or 2 FMTs. Single or optional re-treatment	Pouchoscopy. Delivered to the most proximal point of insertion (proximal pouch or neo-terminal ileum)	250/500 mL in total derived from 25/50 g stool	1/11 after 4 weeks. Only 11 underwent pouchoscopy at 4 weeks	1/11 after 4 weeks	Mean ePDAI of 3.38 at baseline decreased to 3.36 after four weeks	Mean hPDAI of 1.05 at baseline increased to 1.36 after four weeks	One patient admitted to hospital 8 days after FMT because of abdominal pain considered not related to FMT. One patient diagnosed with Crohn’s disease at pouchoscopy after four weeks, which was suspected in advance. Minor self-limiting adverse events. Discomfort (4), flatulence (4), bloating or cramping (3), fatigue (3), and nausea (2).	12 months
Stallmach et al., 2016	Prospective open-label pilot study	Chronic antibiotics resistant pouchitis (three or more cycles of antibiotics)	5	1–7 FMTs	FMT to the jejunum with intervals of 3–4 weeks	275 mL fecal saline suspension derived from 75 g stool	5/5 after last FMT.	4/5 after last FMT	Mean ePDAI of 3.8 at baseline decreased to 1.2 after last FMT	Mean hPDAI of 3 at baseline decreased to 1.2 after last FMT	Mild transient fever and CRP increase in one patient.	3 months. One patient followed for 12 months
Steube et al., 2017	Prospective open-label pilot study	Chronic antibiotics resistant pouchitis	14	2–4 FMTs	Nasojejunal or capsule application delivered every 4 weeks according to treatment outcome	NA	7/14. PDAI scores however not described. Assessed through F-Calprotectin.	NA	NA	NA	No reported adverse events.	8 weeks

NA, not available; cPDAI, clinical Pouchitis Disease Activity Index; ePDAI, endoscopic Pouchitis Disease Activity Index; hPDAI, histologic Pouchitis Disease Activity Index; FMT, Fecal Microbiota Transplantation; RCT, Randomized Controlled Trial; mPDAI, modified Pouchitis Disease Activity Index; F-Calprotectin, Fecal Calprotectin.

**Table 3 microorganisms-08-01433-t003:** Preparation and delivery of FMT material.

Author and Year	Donor(s)	Fresh/Frozen	FMT Preparation	Concomitant Treatment	Pre-Treatment with Antibiotics	Single/Multi-donor	Bowel Cleansing
Fang et al., 2016	Unrelated donor	Fresh	Stool mixed with sterile saline and diluted to 250 mL	The patient was off his usual antibiotics during the entire follow-up period but continued antidiarrheal medication.	Antibiotics stopped 48 h before FMT	Single	No
Herfarth et al., 2019	Unrelated donor with high butyrate production	Frozen	Stool bank provided FMT for endoscopic administration, FMT capsules and matching placebos	NA	Antibiotics stopped 24 h before FMT	Single	NA
Kousgaard et al., 2020	Unrelated	Frozen	20 g stool mixed with 100 mL sterile water, blended and filtered, 10% glycerol	Concurrent therapies were permitted if stable with exception of antibiotics, probiotics, and biologic treatment.	Antibiotics stopped 7 days before FMT	Multi-donor (5 donors)	No
Landy et al., 2015	Relative/partner (6) or anonymous (2)	Fresh (less than 6 h prior to FMT)	30 g of stool was homogenized with a household blender in 50 mL of 0.9% saline and filtered through sterile gauze to produce a fecal-saline solution	Night before the procedure the recipient was treated with a proton pump inhibitor (Omeprazole 20 mg).	NA	Single	NA
Nishida et al., 2019	Second degree relatives	Fresh (less than 4 h prior to FMT)	150–200 g donor stool mixed with 350–500 mL sterile saline and filtered through gaze	NA	NA	Single	Polyethylene glycol solution
Schmid et al., 2017	Patient’s son	Fresh (less than 6 h prior to FMT)	A fresh stool sample diluted with 500 mL saline and filtered through gaze	NA	NA	Single	Enemas
Selvig et al., 2020	Unrelated	Frozen	25 g of stool mixed with saline to a 250 mL fecal suspension	Accepted with antibiotics as exception. Three patients continued biologic treatment.	7/18 patients received 5 days of 550 mg Rifaximin beginning 8 days prior to treatment	Single (from 11 different donors)	Magnesium citrate a day before FMT and phosphate enema at day of FMT
Stallmach et al., 2016	Unrelated	Fresh (first FMT) and frozen	150 g stool mixed with 400 mL saline. Then filtered and separated in two. The first immediately delivered to the patient and the second stored at −80 °C	NA	NA	Single (from 2 different donors)	No
Steube et al., 2017	Unrelated	NA	NA	NA	NA	Single (from 3 different donors)	NA

NA, not available; FMT, Fecal Microbiota Transplantation.

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
