# Peer review of "Fecal Microbiota Transplantation in the Treatment of Chronic Pouchitis: A Systematic Review"

_microorganisms, 2020, doi:10.3390/microorganisms8091433_

Round 1

Reviewer 1 Report

  • Why did you first register your study on PROSPERO and then you did not wait for the result
  • “refractory to  medicinal  1”

Or in case of colorectal cancer

  • “confined to  the  IPAA”

Do you mean to the pouch?

  • “Pouchitis often  develops  as  an  acute,  but  transient,  infection”

Is acute pouchitis an infection??

  • Did you find 669 duplicates?

Author Response

Reviewer: 1

Point 1: Why did you first register your study on PROSPERO and then you did not wait for the result

Response 1: We submitted the protocol 10th of February 2020. Based on the recommendations from PROSPERO we did not wait for approval of the protocol before beginning the work on the systematic review. Further the corona-situation made us fear that we could wait for a very long time for the approval, which we luckily did not, and hence we decided to perform the literature search 15th of April.

Point 2: “refractory to medicinal 1” Or in case of colorectal cancer

Response 2: We agree that this is missing.
P1L41-42: added ”and in some cases of colorectal cancer and familial adenomatous polyposis”

Point 3: “confined to the IPAA” Do you mean to the pouch?
Response 3: We agree that this is confusing and has now changed IPAA to pouch.
P1L43-P2L1: changed to “mainly confined to the pouch”

Point 4: “Pouchitis often develops as an acute, but transient, infection” Is acute pouchitis an infection??
Response 4: We agree that infection is misleading and has now changed the word to inflammation.
P2L2: changed to “Pouchitis often develops as an acute, but transient, inflammation responding to antibiotics.”

Point 5: Did you find 669 duplicates?
Response 5: We only found 174 duplicates. Figure 1 has now been updated to make this more obvious. P4Figure1

Reviewer 2 Report

The presented study has performed a systematic review on FMT treatment in chronic pouchitis patients with an objective to evaluate the clinical outcomes, safety, and different approaches to FMT preparation and delivery. This is the second systematic review study that reports the effects of FMT on symptoms of chronic pouchitis by summarizing available small number of case studies and RCT.

Below are the reviewers' critiques;

In method section, section 2.2 and 2.3 can be combined into one section.

Either in table 2 or table 3, add a column to describe the frequency of FMT treatment,

In the discussion section 4.9, describe whether FMT from IPAA undergone patients with no pouchitis symptoms can be an ideal donor for the chronic pouchitis.

Author Response

Reviewer: 2

The presented study has performed a systematic review on FMT treatment in chronic pouchitis patients with an objective to evaluate the clinical outcomes, safety, and different approaches to FMT preparation and delivery. This is the second systematic review study that reports the effects of FMT on symptoms of chronic pouchitis by summarizing available small number of case studies and RCT.

Below are the reviewers' critiques;

Point 1: In method section, section 2.2 and 2.3 can be combined into one section.

Response 1: P3L15-35: The section 2.2 and 2.3 are now combined into one section.

Point 2: Either in table 2 or table 3, add a column to describe the frequency of FMT treatment,

Response 2: P5Table2: In Table 2, the frequency of the FMT treatment is included in the column “FMT treatment”.

Point 3: In the discussion section 4.9, describe whether FMT from IPAA undergone patients with no pouchitis symptoms can be an ideal donor for the chronic pouchitis.

Response 3: Evidence on the ideal donor for FMT to chronic pouchitis is still lacking, and we agree that perhaps a pouch patient without episodes of pouchitis could be the ideal donor. This has now been added in the discussion section 4.6.

P13L24-25: added “Patients with a pouch without episodes of pouchitis possibly have a healthy microbiome preventing pouchitis and could maybe be ideal donors.”